# Humoral Immune Response in Immunized Sheep with Bovine Coronavirus Glycoproteins Delivered via an Adenoviral Vector

**DOI:** 10.3390/pathogens13070523

**Published:** 2024-06-21

**Authors:** Annamaria Pratelli, Paolo Capozza, Sergio Minesso, Maria Stella Lucente, Francesco Pellegrini, Maria Tempesta, Valentina Franceschi, Canio Buonavoglia, Gaetano Donofrio

**Affiliations:** 1Department of Veterinary Medicine, University of Bari, 70010 Valenzano, Italy; paolo.capozza@uniba.it (P.C.); mariastella.lucente@uniba.it (M.S.L.); francesco.pellegrini@uniba.it (F.P.); maria.tempesta@uniba.it (M.T.); buonavoglia.canio@gmail.com (C.B.); 2Department of Veterinary Science, University of Parma, 43126 Parma, Italy; sergio.minesso@unipr.it (S.M.); valentina.franceschi@unipr.it (V.F.); gaetano.donofrio@unipr.it (G.D.)

**Keywords:** bovine coronavirus, vaccine, immune response

## Abstract

Bovine coronavirus (BCoV) is distributed globally and mainly causes different clinical manifestations: enteric diarrhea in calves, winter dysentery in adults, and respiratory symptoms in cattle of all ages. Low mortality and high morbidity are the hallmarks of BCoV infection, usually associated with substantial economic losses for the livestock industry. Vaccination, combined with the implementation of biosecurity measures, is the key strategy for the prevention of infections. This pilot study evaluates the immunogenicity of a recombinant vaccine containing two BCoV antigens (S and M) in sheep, compared to vaccines containing only the M or S protein. Three groups of sheep were inoculated intramuscularly at day 0 and day 21 with recombinant adenoviruses expressing BCoV S protein (AdV-BCoV-S), BCoV M protein (AdV-BCoV-M), or both proteins (AdV-BCoV-S + M). Serum antibodies were evaluated using immunofluorescence (IF) and serum neutralization (SN) tests. Moderate seroconversion was observed by day 21, but serum antibodies detected via SN increased from 1:27.5 (day 21) to 1:90 (day 28) in sheep inoculated with the recombinant AdV expressing both the S- and M-BCoV proteins. Based on the SN results, a repeated-measures ANOVA test indicated a more significant difference in immune response between the three groups (F = 20.47; *p* < 0.001). The experimental investigation produced satisfactory results, highlighting that the S + M recombinant vaccine was immunogenic, stimulating a valid immune response. Despite some inherent limitations, including a small sample size and the absence of challenge tests, the study demonstrated the efficacy of the immune response induced via the recombinant vaccine containing both S and M proteins compared to that induced via the individual proteins S or M.

## 1. Introduction

Coronaviruses (CoVs) are enveloped, positive, single-stranded RNA viruses affecting animals and humans with the ability to cross interspecies barriers, therefore inducing zoonotic infections [1,2]. Bovine coronavirus (BCoV) belongs to the family *Coronaviridae*, the genus *Betacoronavirus*, and the subgenus *Embecovirus* which includes, among others, the common cold virus of humans (HCoV-OC43) and the viruses associated with the epidemic severe acute respiratory syndrome coronavirus type 1 (SARS-CoV-1) and Middle East respiratory syndrome coronavirus (MERS-CoV), as well as the pandemic SARS-CoV-2 [3,4]. The genome, of about 31 Kb, includes several open reading frames (ORFs) coding for non-structural (nsp) and structural proteins (sp). The major sp are the nucleocapsid protein N, the spike glycoprotein S, the membrane protein M, the hemagglutinin–esterase protein HE, and the small-membrane protein SE [5]. The large heterodimeric glycoprotein S comprises two subunits: the N-terminal domain (NTD) S1, responsible for receptor binding and tissue tropism, and the C-terminal domain (CTD) S2, responsible for fusion. The S protein contains the epitopes that are the major targets for the antibody response, activating the cell-mediated immune response and representing the major site of genetic and antigenic variations among different BCoV strains, which likely contributes to vaccines’ effectiveness [6,7]. The HE protein also interacts with cell membrane receptors, and some epitopes are the targets of neutralizing antibodies, while the type-III M glycoprotein is the most abundant viral sp, and it plays a role in neutralizing viral infectivity [8]. The basic phosphoprotein N protein forms the helic nucleocapsid and modulates RNA synthesis, and SE participates in envelope assembly [9,10].

BCoV is endemic worldwide and is responsible for significant economic losses due to increases in mortality, reduced growth and milk production, more demanding animal management, and veterinary medical expenses [11,12]. The virus, which may be frequently identified in healthy cattle, can affect both respiratory and enteric apparatuses, causing different syndromes such as neonatal diarrhea, “wintery disease” in adults, and respiratory disease as a cofactor of Bovine Respiratory Syndrome (BRD) [5,13,14,15]. Several studies have confirmed that BCoV exists as a “quasispecies” with one serotype, suggesting that differences observed among strains in the antigenic spectrum, tropism, and virulence are not related to their clinical origin but are the expression of complex interactions among the virus–host environment [4,16].

Several domestic and wild ruminants are susceptible to BCoV infection, and CoV strains with genetic and antigenic features of BCoV (“bovine-like CoVs”) have been identified in the feces and respiratory secretions of animals with or without clinical signs [17,18]. These data suggest that cattle may act as reservoirs for CoV strains and may aid in the maintenance and evolution of these viruses among animals and humans [19]. The presence of a common ancestor or, alternatively, an origin from BCoV has been hypothesized for several betacoronaviruses, such as HCoV-OC43, bubaline CoV (BuCoV), equine CoV (ECoV), and canine respiratory coronavirus (CRCoV) [20,21,22], also considering that the risk of interspecies spillover is favored by the presence of precarious hygienic and sanitary conditions, as well as close interaction between animals and between animals and humans.

Currently, three inactivated vaccines are administered to pregnant cows to enhance the antibody-based protection of newborn calves against BCoV and other gastrointestinal infectious diseases that occur in the early stages of life [23,24]. In addition, modified live virus vaccines are orally administered in neonatal calves to stimulate a strong immune response [23,24]. Although BCoV-infected adults do not always exhibit clinical symptoms, they can still transmit the virus, posing a significant challenge in controlling the spread of the disease among newborn calves. Therefore, there is an urgent need for powerful and effective vaccines for cattle, and to improve the overall effectiveness of immunization, it is essential both to overcome the limitations of the currently existing BCoV vaccines and extend their use to older cattle.

Although several epidemiological studies indicate a potential correlation between vaccine-induced antibodies to BCoV and immunological protection, it remains uncertain whether serum antibodies also provide protection against respiratory BCoV diseases and whether their presence simply indicates previous exposure to BCoV [11].

The current focus on the vaccination of young calves leaves a considerable gap in the protection of adult cattle and the general cattle population. Many challenges are associated with the use of BCoV vaccines in cattle, including the development of more effective vaccines, the selection of suitable formulations and adjuvants, and the use of new genetically engineered vaccines that can elicit strong immune responses in all animals [25]. Based on these observations, the study of the immune response of BCoV after vaccination allows for an expansion of our knowledge on BCoV infections and on protective immunity, translating the information into human medicine for the control of SARS-CoV-2 infection, thus fulfilling the “One-Health” approach. Starting from November 2021, the COVID-19 vaccination using adenoviral vector vaccines and mRNA vaccines, both based on the viral S protein as a targeted antigen, has allowed for the control of the pandemic [26]. The aim of the present study was to evaluate whether a vaccine formulated with two BCoV antigens (S plus M) was more immunogenic than one formulated with the S or the M protein alone. To this purpose, the immune response induced via two recombinant adenoviruses expressing the S or the M proteins from BCoV and inoculated intramuscularly (i.m.) was evaluated in sheep.

## 2. Materials and Methods

### 2.1. Cells

HEK (human embryo kidney cells) 293 T (ATCC: CRL-11268) were grown in complete Eagle’s minimal essential medium (cEMEM: 1 mM of sodium pyruvate, 2 mM of L-glutamine, 100 IU/mL of penicillin, 100 μg/mL of streptomycin, and 0.25 μg/mL of amphotericin B), supplemented with 10% FBS, and incubated at 37 °C/5% CO_2_ in a humidified incubator. All the supplements for the culture medium were purchased from Gibco (Gibco, Segrate (MI), Italy). Ovine bone marrow mesenchymal stromal (OvBM-MSc) primary cells were generated from ovine fetal bone marrow, as previously described (https://www.liebertpub.com/doi/10.1089/clo.2005.7.154 (accessed on 1 July 2023)), and maintained in cEMEM with 10% FBS. HEK 293T, and OvBM-MSc were also grown in cEMEM supplemented with 10% Ovine serum (OvS) (Biowest, Nuaillé, France).

### 2.2. Adenovirus Reconstitution

HEK 293 T cells were seeded into 25 cm^2^ flasks (1 × 106 cells/flask) and incubated at 37 °C with 5% CO_2_. When the cells were sub-confluent, the culture medium was removed, and the cells were transfected with pAd5-bovS-ΔRS-HA-GFP, pAd5-bovM-HA-GFP, or pAd5-GFP (mock control) using the Polyethylenimine (PEI) transfection reagent (Polysciences, Inc., Warrington, PA, USA). Briefly, 3 µg of DNA was firstly linearized through PacI (Thermoscientific, Waltham, MA, USA) restriction enzyme digestion and then mixed with 7.5 µg of PEI (1 mg/mL) (ratio: 1:2.5 DNA/PEI) in 500 µL of Dulbecco’s modified essential medium (DMEM) with high glucose (Euroclone) and without serum. After 15 min of incubation at room temperature, 2000 µL of the medium without serum was added, and the transfection solution was transferred to the cells (monolayer) and left for 6 h at 37 °C with 5% CO_2_ in a humidified incubator. The transfection mixture was then replaced with a fresh cEMEM medium, with 10% FBS, and incubated at 37 °C with 5% CO_2_. The cells were checked daily for GFP-expressing plaques via fluorescence microscopy. When ~50% of the cell monolayer was affected by CPE, the flasks were frozen at −80 °C. The flasks were subsequently thawed, and the culture supernatant, containing the recombinant viral particles, was firstly clarified via centrifugation at 3500 rpm for 10 min at 4 °C and then filtered through a 0.45 µm filter (https://www.ncbi.nlm.nih.gov/pmc/articles/PMC44694/ (accessed on 28 August 2023)).

### 2.3. Adenoviral Vector Growth and Titering

HEK293T cells were seeded in 25 cm^2^ flasks (0.5 × 10^6^ cells/flask) and incubated at 37 °C with 5% CO_2_. When the cells were sub-confluent, the culture medium was removed, and the cells were infected with 0.5 mL of the cell supernatant harvested during the adenoviral vector reconstitution in HEK293T cells. When the CPE affected ~50% of the cell monolayer, the flasks were frozen, and after thawing, the clarified and filtered supernatant was titrated in HEK293T cells. Cells were infected with 10-fold dilution of the adenoviral vectors, and after 3 days of culturing, the transducing units (T.U.)/mL were estimated. Once the titer was established, the adenoviral vectors were then amplified, infecting HEK 293T cells at a multiplicity of infection (M.O.I.) of 0.1, and titrated again after the freezing/thawing of the flasks (https://www.ncbi.nlm.nih.gov/pmc/articles/PMC44694/ (accessed on 28 August 2023)). To produce the recombinant viruses for the ovine immunization study, HEK293T cells were infected with 0.1 M.O.I. of Ad5-bovS-ΔRS-HA-GFP (Ad5-S) or Ad5-bovM-HA-GFP (Ad-M) in cEMEM with 10% of OvS. After 3 h of incubation, the viral inoculum was removed from the cells and substituted with fresh cEMEM supplemented with 10% of OvS (Biowest), and the cells were incubated at 37 °C, 5% CO_2_. Forty-eight hours post-infection, the CPE was evident on the cell culture monolayer. The flasks were frozen/thawed, and the culture supernatant, containing the recombinant viral particles, was directly harvested after the clarification of cellular debris. Adenoviruses were classically titrated on HEK293T cells [27,28,29].

### 2.4. MTT Assay

The 3-(4,5-dimethylthiazol-2-yl)-2,5-diphenyltetrazolium bromide (MTT, Sigma-Aldrich (Merck), Tokyo, Japan) cell metabolic assay was used to measure cell viability. Briefly, 3 × 10^3^ HEK293T cells were seeded in 96-well plates, 100 µL/well, in cEMEM with 10% FBS or OvS and incubated at 37 °C, 5% CO_2_. After 24, 48, 72, and 96 h of incubation, the cells were incubated for a further 6 h with 10 µL/well (corresponding to 50 µg/well) of MTT, dissolved in sterile PBS, then 110 µL/well of solubilization solution (10% SDS in HCl 0.01 M) was added and incubated overnight at 37 °C. The optical density was measured at 620 nm in a microplate reader (Multiskan FC, ThermoScientific, Waltham, MA, USA). Statistical differences among treatments were tested via an analysis of variance (ANOVA).

### 2.5. Immunoblotting

A Western immunoblotting analysis was performed on protein cell extracts from 25 cm^2^ flasks of OvBM-MSc infected with 1 M.O.I. of Ad5-bovS-ΔRS-HA-GFP (Ad5-S) or Ad5-bovM-HA-GFP (Ad-M) or left un-infected. For protein extraction, 100 µL of cell extraction buffer (50 mM Tris–HCl, 150 mM NaCl, and 1% NP-40; pH 8) was added to each cell pellet, and total protein quantification was performed using a BCA Protein Assay kit (Pierce™, ThermoScientific). Different amounts of protein samples were electrophoresed on 10% SDS-PAGE and then transferred to PVDF membranes (Millipore, Merck, Rahway, NJ, USA) via electroblotting. The membrane was blocked in 5% skim milk (BD), incubated at 1 h with a primary mouse monoclonal antibody anti-HA tag (G036, Abm Inc., New York, NY, USA), diluted at 1:10,000, and then probed with horseradish peroxidase-labeled anti-mouse immunoglobulin (A9044, Sigma-Aldrich (Merck), Tokyo, Japan), diluted at 1:15,000, and finally visualized using enhanced chemiluminescence (Clarity Max Western ECL substrate, Bio-Rad, Hercules, CA, USA).

### 2.6. Immunofluorescence Staining of Adenovirus-Infected Cells

Sub-confluent monolayers of OvBM-MSc were infected with either Ad5-bovS-ΔRS-HA-GFP or Ad5-bovM-HA-GFP at an M.O.I. of 1TCID_50_. After 24 h, the cells were fixed with 4% paraformaldehyde for 10 min and then permeabilized with 1% Triton-X 100 in PBS for 10 min. Following three washes with PBS, the cells were blocked for 1 h at room temperature with 10% FBS diluted in PBS containing 1% BSA (Sigma). Subsequently, the cells were incubated overnight at 4 °C with anti-HA mouse monoclonal antibody (G036, Abm Inc., Vancouver, BC, Canada) diluted at 1:1000 in PBS containing 1% BSA. The antibody was then removed, and the cells were washed extensively with PBS. Next, the cells were incubated with the secondary antibody AlexaFluor 594-conjugated goat anti-mouse IgG (A11032, Life Technologies, Carlsbad, CA, USA) diluted at 1:500 in PBS containing 1% BSA for 1 h at room temperature in the dark. After three washes with PBS, the cells were observed using inverted fluorescence microscopy (Zeiss-Axiovert-S100, Zeiss, Oberkochen, Germany), and images were acquired with a digital camera (Zeiss-Axiocam-MRC, Zeiss, Oberkochen, Germany).

### 2.7. Animals and Experimental Protocol

The experimental study was performed according to animal health and well-being regulations and was authorized by the Ministry of Health of Italy (Authorization n°: 642/2022-PR). Ten young adult sheep (>9 months), three males and seven females, sero- and virus-negative to BCoV, were housed at the “Infectious Disease Unit” of the Animal Hospital at the Department of Veterinary Medicine of University Aldo Moro of Bari, Italy. All the experiments were carried out in strict accordance with the recommendations in the guidelines of the Code for Methods and Welfare Considerations in Behavioural Research with Animals (Directive 86/609EC; RD1201/2005), and all efforts were made to minimize suffering. The procedures were conducted by qualified personnel, and the management of the animals took place under the supervision of veterinary medical experimenters and the veterinarian responsible for Legislative Decree 26/2014. After arrival, the animals were monitored daily for a period of 15 days for their general health status prior to the beginning of the experiment. At day 0, the animals were tested again for BCoV antibodies with an IFA and SN test, randomly grouped, and inoculated i.m. with the AdV-recombinant BCoV proteins as follows:AdV-BCoV-S protein group: three sheep, one male and two female (#1, #2, and #3), were inoculated with 10^8^ infectious units (IU) of AdV-BCoV-S protein.AdV-BCoV-M protein group: three sheep, one male and two female (#4, #5, and #6), were inoculated with10^8^ IU of AdV-BCoV-M protein.AdV-BCoV-S + M proteins group: four sheep, one male and three female (#7, #8, #9, and #10), were inoculated with 10^8^ IU of AdV-BCoV-S protein and 10^8^ IU of AdV-BCoV-M protein simultaneously.

As a control, five sheep housed at the Animal Hospital of the Veterinary Medicine Campus of University Aldo Moro of Bari for didactic purposes were monitored for clinical evaluation.

After vaccine inoculation (day 0), rectal temperatures were recorded from day 0 to day 4. One booster inoculation with the same amount of vaccine was performed following the same schedule after 21 days (day 21), and rectal temperatures were again recorded from day 21 to day 25. Blood samples for serum collection were taken from the jugular vein in Venojet glass tubes at day 0 and day 21 (pre-booster) and at day 28 and day 72 (post-booster). The samples were allowed to clot overnight at +4 °C; then, the serum was obtained via centrifugation at 1500 rpm for 10 min at +4 °C. Serum aliquots were stored at −20 °C until use.

All the animals were examined daily for clinical signs throughout the study to monitor animal health and welfare and ensure that any animal in distress could receive appropriate veterinary care in accordance with standard veterinary practice.

### 2.8. Serological Assays

Serial dilutions of each serum sample were tested for the detection of antibodies to BCoV using the immunofluorescence (IF) and seroneutralization (SN) tests.

(i)IF test. The IF test was performed using Madin Darby Bovine Kidney (MDBK) cells infected with an in vitro-adapted BCoV strain 438/06 [20], deposited on multispot slides, and acetone-fixed. Serial two-fold dilutions of each serum in phosphate-buffered saline (PBS; pH 7.0) starting from 1:10 were tested with anti-bovine IgG fluorescein-conjugated serum (Sigma Chemicals, St. Louis, MO, USA), and the multispot slides were read using a fluorescence microscope.(ii)SN test. SN was carried out in 96-well microtiter plates containing a 24 h monolayer of MDBK cells. Serial two-fold dilutions of each serum in PBS, pH 7.0, starting from 1:5 were mixed with 100 Tissue Culture Infectious Doses (TCIDs)_50_ of BCoV field strain 438/06 titrated via the IF test. After a 1 h incubation at room temperature, each serum/virus mixture was added to each well. After three days of incubation at 37 °C in a CO_2_ incubator, the plates were subjected to the IF test and observed via an inverted fluorescence microscope. The antibody titer was expressed as the highest serum dilution neutralizing the virus (absence of fluorescence).

### 2.9. Data Analyses

All statistical analyses of the variables were performed using the software R version 4.2.2 (R Foundation for Statistical Computing, Vienna, Austria; https://www.R-project.org/ (accessed on 5 February 2024)). Descriptive statistical analyses were applied to characterize the samples. Subjects were grouped according to the typology of vaccine they were inoculated with to improve the understanding of the results. Repeated-measures ANOVA statistics were used to evaluate the differences between the immune responses of the three different vaccine groups (AdV-BCoV-S protein group, AdV-BCoV-M protein group, and AdV-BCoV-S + M proteins group) at different time points (day 0, day 21, day 28, and day 72). The typology of the vaccine inoculated was considered the inter-subject factor, and the time point was considered the intra-subject factor. The Bonferroni statistic was used for intergroup and intragroup multiple comparisons.

## 3. Results

Replication-defective recombinant human adenoviruses type 5, Ad5-bocS-ΔRS-HA-GFP and Ad5-bovM-HA-GFP delivering BCoV S (bovS), and M glycoprotein (bovM) were constructed. Initially, an ORF encoding bovS was designed in silico. The bovS open reading frame (ORF) sequence, derived from a field isolate of the BCoV genome, was depleted of its last 19 bp, potentially coding for a potential endoplasmic reticulum retrieval signal (ERRS; peptide: CCDDYTGHQELVIKTSHDD), and substituted with a hemagglutinin (HA) tag to be monitored in terms of expression via Western immunoblotting. The so-designed ORF, bovS-ΔRS-HA, was human codon usage-adapted with the Jcat codon adaptation tool (http://www.jcat.de (accessed on 1 June 2023)) and chemically synthesized. Similarly, bovM ORF was human codon usage-adapted, HA-tagged to its carboxyterminal (M-HA), and chemically synthesized as for bovS-ΔRS-HA (sequences in Appendix A). Next, each of the two ORFs was integrated into a replicating incompetent type-5 Adeno viral-transfer vector (Ad5) backbone under the control of the CMV immediate early promoter and a bovine growth hormone polyadenylation signal. A GFP expression cassette, to follow viral replication and transduction, was provided, too (Figure 1A). Recombinant Ad5-bovS-ΔRS-HA-GFP and Ad5-bovM-HA-GFP infectious viral particles were reconstituted in HEK293T cells, as shown in their replication competence inducing progressive CPE (Figure 1B,D) and the increase in the viral titer during the post-transfection time (Figure 1C,E). Whereas transducing competence, as well as replication incompetence, was evaluated using ovine cells defective in E1A and E1B genes, in fact, these cells were well transduced in the absence of CPE, as shown in the GFP, bovS, and bovM expression (Figure 1F,G).

Since Ad5-S-ΔRS-HA-GFP and Ad5-M-HA-GFP are generated in HEK293T cells with the presence of FBS, whereas we wanted to immunize ovine, we considered the generation of a protocol to produce Ad5-S-ΔRS-HA-GFP and Ad5-M-HA-GFP free of bovine antigens and avoiding the viral purification step, which is time- and cost-consuming. Therefore, HEK293T cells were adapted to grow with 10% ovine serum (OvS), instead of FBS, and either no appreciable detrimental effect on cellular growth (Figure 2A) or a reduction in the viral vector titer (Figure 2B,C) was observed.

To evaluate the immune response of sheep against BCoV, the three groups of animals were inoculated i.m. at day 0 and day 21 with recombinant AdVs expressing BCoV-S protein (AdV-BCoV-S protein group), BCoV-M (AdV-BCoV-M protein group), both BCoV S and M proteins (AdV-BCoV-S + M proteins group).

The tested animals were always healthy throughout the observation period before the vaccination trials. The temperature remained in the optimal range, the appetite was preserved, and no signs of pain or discomfort were recorded. The animals, confirmed to be serologically negative for BCoV at day 0 via SN (mean value < 1:5) and IFA (mean value < 1:10), were also adequately monitored throughout the course of the study and received the necessary care to guarantee the reliability of our research; their physiology was respected, and for their well-being, unnecessary pain, suffering, and permanent damage were avoided. No side and/or adverse effects were observed in the treated animals except for a feverish rise (from 39.5 °C to 40.6 °C) after the first inoculation (day 0) and for 24 h, detected in 2 animals of AdV-BCoV-M protein group (#4, #5), and in all the four sheep of AdV-BCoV-S + M proteins group (Table 1).

After the booster inoculation at day 21, only two sheep of the AdV-BCoV-M protein group (#4 and #5) and one sheep of the AdV-BCoV-S + M proteins group (#10) showed a fever rise (from 39.7 °C to 40.3 °C) for 24 h (Table 1).

The collected sera were tested for antibodies to BCoV via IF and SN tests. Using an IF test, serum antibodies of all three groups of sheep showed a moderate increase (mean value: 1:40) at day 21 (pre-booster). The antibody titers increased at day 28, one week after the booster inoculation, and reached the highest titers in sheep of the AdV-BCoV-M protein group (mean value: 1:320) compared to the AdV-BCoV-S protein group and the AdV-BCoV-S + M proteins group (mean value: 1:240 and 1:200, respectively). After two and half months (day 72), the mean values of the antibody’s titers were 1:33.3 in sheep of the AdV-BCoV-S protein group and the AdV-BCoV-M protein group, versus 1:120 in sheep of the AdV-BCoV-S + M proteins group (Figure 3).

Using the SN test, antibody titers showed analogous kinetics in the animals of the AdV-BCoV-S protein group and AdV-BCoV-M protein group, but with slightly lower values. SN antibody titers (mean value) increased from 1:26.6 (day 21) to 1:46.6 (day 28) in the sheep of the AdV-BCoV-S protein group and from 1:20 (day 21) to 1:53.3 (day 28) in the animals of the AdV-BCoV-M protein group. In the AdV-BCoV-S + M proteins group, SN antibody titers increased from 1:27.5 (day 21) to 1:90 (day 28). The SN antibodies detected at day 72 showed a sharp decline in all three groups (mean values: 1:13.3, 1:10, and 1:15, respectively) (Figure 3).

We conducted a repeated-measures ANOVA test, considering all IF and SN test results. Mauchly’s test of sphericity indicated that the assumption of sphericity had been violated (χ^2^(5) = 30.699, *p* < 0.001, and χ^2^(5) = 35.886, *p* < 0.001, respectively, for the IF and SN results), and therefore, a Greenhouse–Geisser correlation was used (ε = 0.451; ε = 0.39). The repeated-measures ANOVA test indicated a significant difference in the immune response between the three vaccine groups in the IF results (F = 7.66; *p* = 0.012). Based on the SN results, the repeated-measures ANOVA test indicated a more significant difference in the immune response between the three groups (F = 20.47; *p* < 0.001). In both cases, the post hoc paired *t*-test test using a Bonferroni-corrected α = 0.0083 indicated that the means of the following pairs are significantly different.

## 4. Discussion and Conclusions

BCoV is widely spread in cattle herds worldwide and is responsible for respiratory and enteric diseases in calves that compromise animals’ welfare and cause huge economic losses to farmers [11,12]. The evolution of the infection in cattle is generally acute with an extremely variable clinical pattern and frequent asymptomatic development [14]. Data on the immune response induced via BCoV and on the effectiveness of vaccination prophylaxis are fragmentary and often conflicting [11]. The present study aimed to provide useful information on the immune response to CoVs in animals after vaccination from a translational perspective. In particular, the main objective was to evaluate whether the use of a vaccine containing two CoV proteins (S and M) was more immunogenic than a vaccine prepared with the S protein alone, such as the current vaccines for SARS-CoV-2. To this purpose, the experimental investigation tested the immunogenicity of recombinant adenoviruses expressing BCoV-S and BCoV-M proteins inoculated i.m. in sheep, separated or mixed, through the evaluation of the antibody response [30,31].

The experimental study provided satisfactory results, and the tested vaccines were proven to be immunogenic since a good immune response in the inoculated sheep was induced. Considering the negative antibody titers detected at day 0 in the ten experimental animals and in the sheep that tested seronegative during serological investigations carried out in our department (personal data), the results of the present study highlighted that the sheep inoculated with the recombinant AdVs expressing both S- and M-BCoV proteins developed a valid serological response, especially after the booster inoculation. All four sheep developed neutralizing antibodies (mean value: 1:90) at day 28, most likely highlighting that the combination of the S and M antigens is more immunogenic than the individual administration of the antigen.

Although these results are preliminary, they are supported through statistical analyses. Indeed, a statistically significant distinction in the immune response was detected among the three vaccine groups, as evidenced by the outcomes of the SN and IF tests. Nevertheless, it is critical to recognize that the data violated the assumption of sphericity, as demonstrated via Mauchly’s test. To rectify this error, a Greenhouse–Geisser correlation corrector was implemented. In relation to the outcomes of the IF examination, a statistically significant distinction was observed (F = 7.66; *p* = 0.012); conversely, the disparity was even more pronounced in the serum neutralization (SN) test results (F = 20.47; *p* < 0.001). This implies that the significance of the differentiated immune responses observed among vaccine groups might be impacted due to the measure employed to assess the immune response.

The limitations of our investigation are mainly two-fold: first, the absence of negative controls (due to the need to reduce the overall number of animals used in the experiments), and second, the lack of a reference vaccine that would have facilitated the comparability of the results. Despite the aforementioned limitations, the results presented in this study could represent an important starting point for the development of combined vaccines against CoV infections to be applied not only in the veterinary field but also in the human field.

In our opinion, the most relevant aspect that emerges is the ability of BCoV S and M proteins to stimulate a valid and more efficient immune response when co-inoculated, suggesting comparative studies to be planned for the control of SARS-CoV-2 infection. Our data, although they require further investigations and must be confirmed and re-evaluated with a greater number of animals, open up interesting scenarios for the development of new vaccines for SARS-CoV-2, considering that the vaccine currently used in humans is set up exclusively with the S antigen [32,33].

Undoubtedly, to confirm the immunogenicity of this vaccine and its greater ability to protect against infection and/or disease compared to that induced via vaccines prepared only with the S protein, it will be necessary to complete the study with new experimental investigations and infection trials. Challenge tests, however, currently represent an obstacle to the advancement of studies on the efficacy of these vaccines, as an animal model for BCoV is not currently available. Despite all the limitations of the highlighted study, the study confirmed that the inoculation of S and M proteins stimulates and induces a better immune response than that induced via a single protein (S or M).

## Figures and Tables

**Figure 1 pathogens-13-00523-f001:**
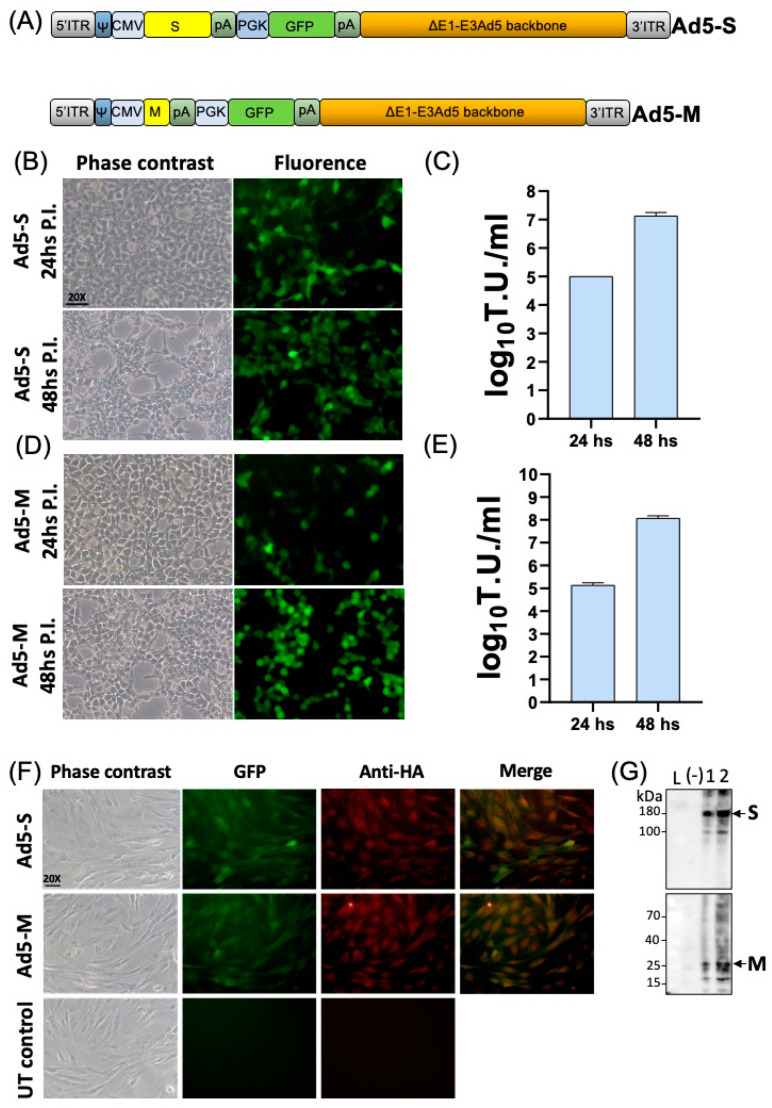
**Construction of an assessment of recombinant adenoviruses expressing BCoV S and M glycoproteins.** (**A**) Diagram (not to scale) of Ad5-bocS-ΔRS-HA-GFP (Ad5-S) and Ad5-bovM-HA-GFP (Ad5-M) genomic structure composed of different elements: 5′ and 3′ inverted terminal repeat (ITR; gray), adenoviral packaging signal (Ψ; dark blue), human cytomegalovirus immediate early enhancer/promoter (CMV; light blue), open reading frame coding for S or M glycoprotein (S or M; yellow), bovine growth hormone polyadenylation signal (pA; dark green), human phosphoglycerate kinase 1 promoter (PGK; light blue), open reading frame coding for green fluorescent protein (GFP; green), Herpes Simplex Virus thymidine kinase polyadenylation signal (pA; dark green), and E1A/B and E3 deleted human adenovirus type 5 genome backbone (ΔE1-E3Ad5 backbone; orange). Phase contrast and fluorescence images (scale bar corresponds to 50 µm) of Ad5-S (**B**) and Ad5-M (**D**) infected HEK293T cells at 24 and 48 h (hs) post-infection (P.I.). Ad5-S and Ad5-M viral titer, (**C**) and (**E**), respectively, was measured and expressed as log10 per ml of transducing units (T.U.) of viral particles released at 24 and 48 h P.I. when HEK293T cells were infected with an M.O.I. Values are the means ± standard errors of three independent experiments. (**F**) Immunofluorescent staining of Ad5-S or Ad5-M transduced primary cultures of bone marrow-derived ovine mesenchyme stem/stromal cells and the untransduced control (UT control). Since S and M were HA-tagged, immunostaining was performed with an anti-HA mAb. GFP and anti-HA were colocalized by merging the images (merge). (**G**) Western immunoblotting of Ad5-S or Ad5-M transduced primary cultures of bone marrow-derived ovine mesenchyme stem/stromal cells. Two different number of cell protein extracts were loaded (1 = 10 µg, and 2 = 30 µg), the negative control (−) was made with 30 µg of protein extract that came from untransduced primary cultures of bone marrow-derived ovine mesenchyme stem/stromal cells, and L is the protein-size ladder lane (kDa).

**Figure 2 pathogens-13-00523-f002:**
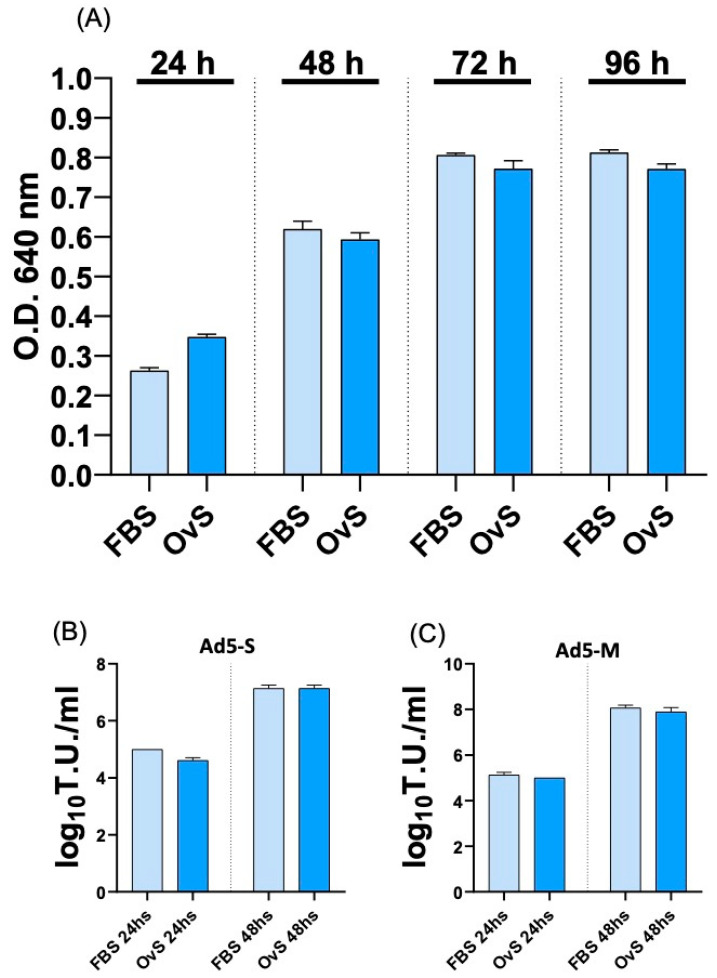
**Assessment of HEK293T growth in OvS.** MTT assay comparing HEK293T growth with medium containing 10% of FBS and the same number of HEK293T growth with medium containing 10% of OvS at different time points (24, 48, 72, and 96 h); absolute values are expressed as optical density (O.D.) (**A**) The data presented are the means ± standard errors of triplicate measurements (*p* > 0.05, as measured with Student’s *t* test). (**B**,**C**) Ad5-S and Ad5-M titer was measured and compared between cells’ growth with FBS or OvS at 24 and 48 h post-infection and expressed as log_10_ T.U. per mL of viral particles released. Values are the means ± standard errors of three independent experiments.

**Figure 3 pathogens-13-00523-f003:**
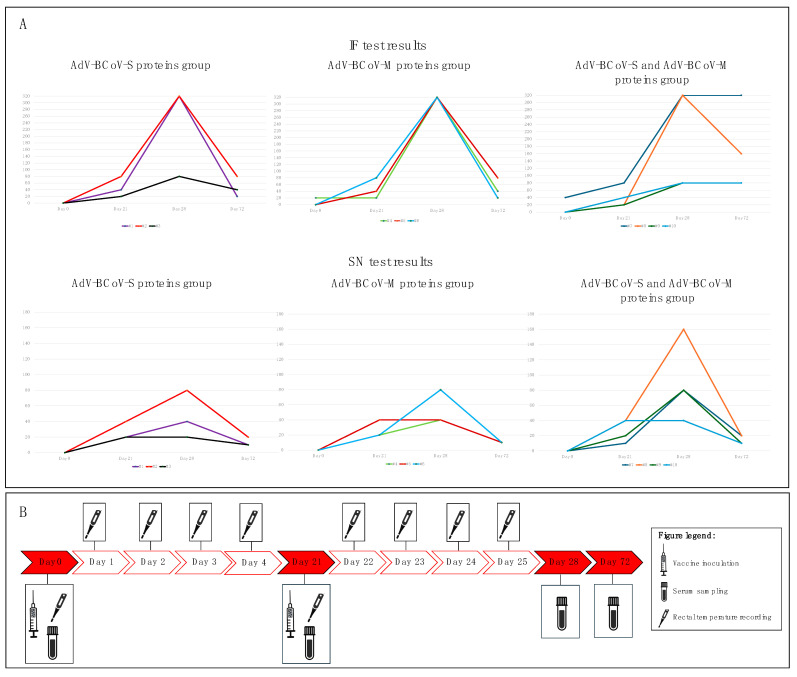
Monitoring of immune response in the AdV-BCoV-S protein group, AdV-BCoV-M protein group, and AdV-BCoV-S + M proteins group using IF test and SN test. In the plots, the x-axis represents the time point (days) in which the antibody titer was evaluated, while the y-axis represents the antibody titer expressed as an absolute value. (**A**): The antibody titer detected in each animal of the three groups included in the experiment using IF and SN at four checkpoints (day 0, day 21, day 28, and day 72). (**B**): Timeline of the procedure conducted during the experiment.

**Table 1 pathogens-13-00523-t001:** Rectal temperatures recorded from day 0 to day 4 and from day 21 (booster day) to day 25. Hyperthermia compared 24 h post-inoculation (day 1) in two animals of the AdV-BCoV-M protein group and in all the animals of the AdV-BCoV-S + M proteins group. After the booster inoculation at day 21, two sheep of the AdV-BCoV-M protein group and one sheep of the AdV-BCoV-S + M proteins group showed a fever rise at day 22. Parameters exceeding the normal ranges are in red.

	Vaccine *	Day of Rectal Temperature Detection									
		0	1	2	3	4	21	22	23	24	25
#1 ^§^	A	38.7	39.4	39	39.2	39.3	39.2	39.2	39	39.0	38.7
#2 ^§^		38.7	39.3	39	39.1	39.2	38.8	39.5	38.7	38.7	38.7
#3 ^§^		38.3	38.5	38.7	38.8	39.1	38.2	39.0	38.6	38.4	38.5
#4 ^§^	B	39.3	40.0	38.7	39.0	39.2	39.0	40.3	38.7	39.2	38.9
#5 ^§^		39.2	40.3	38.7	38.8	38.9	38.7	39.7	38.7	38.9	38.7
#6 ^§^		38.9	38.9	38.9	38.9	39.2	39.0	39.1	38.7	38.9	38.9
#7 ^§^	C	39.2	39.5	38.9	38.9	38.9	38.6	38.7	38.2	38.7	38.2
#8 ^§^		39.1	39.5	39.0	39.2	39.2	38.8	38.8	38.7	38.9	39.0
#9 ^§^		39.1	40.1	38.7	39.2	38.8	38.8	38.5	38.4	38.4	38.2
#10 ^§^		39.5	40.6	39.2	39.5	39.2	38.8	39.9	38.7	39.0	38.7

^§^ Sheep included in the vaccinal trial. * A: AdV-BCoV-S protein. B: AdV-BCoV-M protein. C: AdV-BCoV-S + M proteins.

## Data Availability

The raw data will be made available upon request.

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
