# Peer review of "Humoral Immune Response in Immunized Sheep with Bovine Coronavirus Glycoproteins Delivered via an Adenoviral Vector"

_pathogens, 2024, doi:10.3390/pathogens13070523_

Round 1
Reviewer 1 Report
Comments and Suggestions for Authors
The manuscript titled "Comparative humoral immune response of recombinant Adenoviruses expressing the bovine coronavirus (BCoV) S and M glycoproteins in sheep" evaluates the efficacy of a vaccine containing two BCoV antigens (S and M) compared to those containing only the M or S protein using sheep as a model organism. After careful reading, this reviewer has following suggestions for the authors:
1. Abstract can be improved; needs more information regarding data.
2. Introduction can be significantly improved, highlighting the literature reporting on the immunogenicity of related vaccines from the past.
3. Methods: 2.7: Animals and experimental protocol - What was the age of sheep involved in the study?
4. Lines 357- 360: "Authors should discuss ...... may also be highlighted". These sentences can be deleted.
5. At certain places in Discussion section, where claims are made from the literature, the citations should be provided, which are currently missing. For example, Lines 361-363 and many other places.
6. Discussion can be improved; also more citations needed in Discussion section.
Comments on the Quality of English LanguageEnglish is overall fine; minor corrections needed, in subscripts or superscripts.
Author Response
Dear Reviewer 1,
thank you for giving us the opportunity to submit a revised draft of our manuscript titled: “Comparative humoral immune response of recombinant Adenoviruses Expressing the bovine coronavirus (BCoV) S and M glycoproteins in sheep”, Manuscript Number: pathogens-3039664, to Pathogens.
We appreciate the time and effort that you and the reviewers have dedicated to providing your valuable feedback on our manuscript. We are grateful to the reviewers for their insightful comments on our paper. We have been able to incorporate changes to reflect the suggestions provided by the editor and reviewers. We have highlighted the changes within the manuscript.
Here is a point-by-point response to your comments and concerns.
Comments from Reviewer 1 (R1)
The manuscript titled "Comparative humoral immune response of recombinant Adenoviruses expressing the bovine coronavirus (BCoV) S and M glycoproteins in sheep" evaluates the efficacy of a vaccine containing two BCoV antigens (S and M) compared to those containing only the M or S protein using sheep as a model organism. After careful reading, this reviewer has following suggestions for the authors:
R1.1: Abstract can be improved; needs more information regarding data.
Reply to R1.1: Thank you for your feedback. According to the reviewer’s suggestion we revise the Abstract to include more detailed information regarding the data.
R1.2: Introduction can be significantly improved, highlighting the literature reporting on the immunogenicity of related vaccines from the past.
Reply to R1.2: Following the referee's assessment, we have included in the introduction more information regarding the immunogenicity of related vaccines from the past.
R1.3: Methods: 2.7: Animals and experimental protocol - What was the age of sheep involved in the study?
Reply to R1.3: All subjects included in the experiment were young adult sheep (>9 months).
R1.4: Lines 357- 360: "Authors should discuss ...... may also be highlighted". These sentences can be deleted.
Reply to R1.4: According to the reviewer suggestion we deleted the sentence in the modified manuscript.
R1.5: At certain places in Discussion section, where claims are made from the literature, the citations should be provided, which are currently missing. For example, Lines 361-363 and many other places.
Reply to R1.5: After carefully checking, we inserted several bibliographical references, as the reviewer suggested.
R1.6: Discussion can be improved; also more citations needed in Discussion section.
Reply to R1.6: Thank you for your valuable feedback. The Discussion section was improved and more citations added to support the arguments presented.
Comments on the Quality of English Language: English is overall fine; minor corrections needed, in subscripts or superscripts.
Reply: The manuscript has been extensively revised for the use of the English language.
Reviewer 2 Report
Comments and Suggestions for Authors
This manuscript presents promising initial findings on the immunogenicity of a dual S and M protein vaccine against Bovine Coronavirus (BCoV). However, several limitations and areas for improvement need to be addressed before publication.
- The introduction lacks sufficient background on currently available BCoV vaccines. Provide a brief overview of existing vaccination strategies and their limitations to better highlight the novelty and potential advantages of the proposed dual S and M protein vaccine.
- The design and construction of the Adenovirus vector method used in the study should be described in detail in the Methods section. This would allow other researchers to replicate your work more accurately.
- The current title does not accurately reflect the study's focus. Consider revising it to clearly state the use of a dual S and M protein vaccine delivered via an Adenovirus vector and its impact on the immune response.
- Methods (Section 2.6): Replace "Immunostaining" with "Immunofluorescence staining of adenovirus-infected cells" for improved clarity.
- Experimental Timeline: for better clarity and consistency, replace the generic "T" with specific time points (days) throughout the manuscript when describing the experimental timeline.
- Figure 1: : It appears that only the magnification of the objective lens is provided. Including the magnification level of the camera lens and incorporating a scale bar would enhance the interpretability of the images.
- Figure 3: Improve clarity by using actual days instead of generalized time points for the x-axis. Ensure both the x and y-axis labels are clearly defined in the legend.
- The manuscript lacks data on negative controls and a commercial BCoV vaccine control. Including this information is crucial for interpreting the experimental results and establishing a baseline for comparison.
By addressing these comments, the authors can significantly strengthen the manuscript and enhance its contribution to the field of BCoV vaccines.
Comments on the Quality of English LanguageThere are some typos or errors in the manuscript.
Author Response
Dear Reviewer 2,
thank you for giving us the opportunity to submit a revised draft of our manuscript titled: “Comparative humoral immune response of recombinant Adenoviruses Expressing the bovine coronavirus (BCoV) S and M glycoproteins in sheep”, Manuscript Number: pathogens-3039664, to Pathogens.
We appreciate the time and effort that you and the reviewers have dedicated to providing your valuable feedback on our manuscript. We are grateful to the reviewers for their insightful comments on our paper. We have been able to incorporate changes to reflect the suggestions provided by the editor and reviewers. We have highlighted the changes within the manuscript.
Here is a point-by-point response to your comments and concerns.
Comments from Reviewer 2 (R2)
This manuscript presents promising initial findings on the immunogenicity of a dual S and M protein vaccine against Bovine Coronavirus (BCoV). However, several limitations and areas for improvement need to be addressed before publication.
R2.1: The introduction lacks sufficient background on currently available BCoV vaccines. Provide a brief overview of existing vaccination strategies and their limitations to better highlight the novelty and potential advantages of the proposed dual S and M protein vaccine.
Reply to R2.1: Following the referee's assessment, we have included in the introduction more information regarding the background on currently available BCoV vaccines.
R2.2: The design and construction of the Adenovirus vector method used in the study should be described in detail in the Methods section. This would allow other researchers to replicate your work more accurately.
Reply to R2.2: Ad-5-based vector is a very well-established platform therefore, we decided to introduce some of citations.
He, T.C., Zhou, S., da Costa, L.T., Yu, J., Kinzler, K.W., and Vogelstein, B. (1998). A simplified system for generating recombinant adenoviruses. Proc Natl Acad Sci U S A 95(5), 2509-2514. doi: 10.1073/pnas.95.5.2509.
Bett, A.J., Haddara, W., Prevec, L., and Graham, F.L. (1994). An efficient and flexible system for construction of adenovirus vectors with insertions or deletions in early regions 1 and 3. Proc Natl Acad Sci U S A 91(19), 8802-8806. doi: 10.1073/pnas.91.19.8802.
Duigou, G.J., and Young, C.S. (2005). Replication-competent adenovirus formation in 293 cells: the recombination-based rate is influenced by structure and location of the transgene cassette and not increased by overproduction of HsRad51, Rad51-interacting, or E2F family proteins. J Virol 79(9), 5437-5444. doi: 10.1128/JVI.79.9.5437-5444.2005
R2.3: The current title does not accurately reflect the study's focus. Consider revising it to clearly state the use of a dual S and M protein vaccine delivered via an Adenovirus vector and its impact on the immune response.
Reply to R2.3: Following the referee's assessment, we have retitled the manuscript.
R2.4: Methods (Section 2.6): Replace "Immunostaining" with "Immunofluorescence staining of adenovirus-infected cells" for improved clarity.
Reply to R2.4: The Reviewer is right, and the sentence might be overstated. To avoid misunderstanding we replace “Immunostaining" with "Immunofluorescence staining of adenovirus-infected cells".
R2.5: Experimental Timeline: for better clarity and consistency, replace the generic "T" with specific time points (days) throughout the manuscript when describing the experimental timeline.
Reply to R2.5: We appreciate your insightful input. We value your proposal to replace the generic "T" with precise time markers (days) throughout the book. This modification will certainly improve the clarity and uniformity in explaining the experimental timeframe. We shall do the necessary modifications to enhance the manuscript's precision and improve its comprehensibility.
R2.6: Figure 1: It appears that only the magnification of the objective lens is provided. Including the magnification level of the camera lens and incorporating a scale bar would enhance the interpretability of the images.
Reply to R2.6: Figure 1 has been corrected as you required.
R2.7: Figure 3: Improve clarity by using actual days instead of generalized time points for the x-axis. Ensure both the x and y-axis labels are clearly defined in the legend.
Reply to R2.7: Following the comments of reviewer 2, we modified the Figure 3 in the revised version of the manuscript. we are aware that the IF and SN results should be expressed as doubling dilutions (e.g. 1:30; 1:180) but by convention absolute values ​​are used for the graphic representation of these data. specifically, specifically, a titer of 1:20 corresponds to 20, 1:40 to 40 and so on.
R2.8: The manuscript lacks data on negative controls and a commercial BCoV vaccine control. Including this information is crucial for interpreting the experimental results and establishing a baseline for comparison.
Reply to R2.8: We appreciate your input. Negative controls were not included in the text because in this case they would have been included in the calculation of the animals used in the experimentation. The research project was authorized by the Italian Ministry of Health and the authorization was granted for only 10 animals. Considering that the project does not include challenge tests, we preferred to use all the 10 animals (divided into 3 groups) for the immunization trials. If we had to include the fourth group as a negative control, we should have reduced the number of animals in the vaccine experimental groups. Regarding the commercial BCoV vaccine controls, this datum is out of the purpose of the work. We wanted just to know if the addiction of a second antigen, gM, could potentially induce an increase of immunity. Which seems to be the case in our experimental setting, thus representing a valuable suggestion for a future vaccine formulation.
By addressing these comments, the authors can significantly strengthen the manuscript and enhance its contribution to the field of BCoV vaccines.
Reply: We appreciate your insightful input. Our objective is not to create a vaccine specifically for cows but to use the cow as a model for future applications in developing a vaccine for SARS-CoV-19 in humans. We value the recommendations and will evaluate how they might augment our manuscript's impact on the broader domain of vaccine research.
Comments on the Quality of English Language: There are some typos or errors in the manuscript.
Reply: The manuscript has been extensively revised for the use of the English language.
Round 2
Reviewer 2 Report
Comments and Suggestions for Authors
The authors carefully revised their manuscript in response to the reviewers' comments. Despite the limitations inherent in the design of the animal study, the explanation provided is deemed adequate. After reviewing the revisions and clarifications, I recommend that the manuscript be accepted for publication in the journal Pathogen.